# Hypomorphic mutations in *ura6* confer 5-FOA resistance in fission yeast

**Constance Kowal**[1,2
, **Ying Liu**[1,2,3
], **Claire Denis**[1], **Benoît Arcangioli**[1,2],
**Stefania Francesconi**[1,2], **Serge Gangloff**[1,2]*

**1** Institut Pasteur, CNRS UMR 3525, Université Paris Cité, Genome Dynamics Unit, Paris, France,
**2** Sorbonne Université, ED515, Paris, France, **3** Changchun Institute of Biological Products Co., Ltd,
Changchun, China

☯ These authors contributed equally to this work.
* serge.gangloff@pasteur.fr

## Abstract

Genome integrity is essential for cellular survival and adaptation across diverse phys-
iological states. The fission yeast *Schizosaccharomyces pombe* relies on conserved
DNA repair pathways to maintain genome stability during proliferative growth and
in the absence of cell division (quiescence/G0). Using 5-fluoroorotic acid (5-FOA)
counter-selection, we examined spontaneous mutation accumulation in both condi-
tions in a wild-type prototrophic strain. Unexpectedly, we identified in growing and
quiescent cells a class of 5-FOA–resistant mutants that, unlike canonical *ura4* or *ura5*
loss-of-function mutants, retain the ability to grow without uracil supplementation.
Genetic analyses showed that this phenotype is stable and segregates as a single
locus. Whole-genome sequencing of tetrads from independent crosses revealed
multiple hypomorphic alleles of *ura6*, which encodes the essential uridylate kinase.
These alleles, comprising non-synonymous substitutions and an in-frame duplication,
cluster within conserved regions of the protein and likely reduce production of the
toxic 5-FOA–derived metabolite while preserving sufficient uracil biosynthesis.

## Introduction

Genome stability is essential for the survival and long-term adaptability of all organ-
isms. In *Schizosaccharomyces pombe*, a well-established model for studying eukary-
otic genome maintenance, mechanisms that preserve genome integrity operate
across both proliferative growth and quiescence induced by nitrogen starvation
[1–3]. During exponential growth, cells must accurately replicate and segregate their
genetic material, while in quiescence cells arrest in G0 phase and rely on efficient
DNA repair and chromatin maintenance pathways to preserve genome integrity over
extended periods [4,5]. Failures in these processes can lead to mutagenesis, chro-
mosomal rearrangements, or loss of viability upon re-entry into the cell cycle.

Health Sciences Center: Tulane University
School of Medicine, UNITED STATES OF
AMERICA

**Peer Review History:** PLOS recognizes the
benefits of transparency in the peer review
process; therefore, we enable the publication
of all of the content of peer review and
author responses alongside final, published
articles. The editorial history of this article is
available here: https://doi.org/10.1371/journal.
pone.0344121

**Data availability statement:** The data are held in a public repository. All 32 files are available from the NIH NLM NCBI database (Bioproject=PRJNA1272799) accessible at: https://www.ncbi.nlm.nih.gov/ All 32 samples can be downloaded at: https://www.ncbi.nlm.nih.gov/biosample?LinkName=bioproject_biosample_all&from_uid=1272799.

**Funding:** The author(s) received no specific funding for this work.

**Competing interests:** The authors have declared that no competing interests exist.

To address mutagenesis, 5-fluoroorotic acid (5-FOA) is a widely used counter-selective agent in fungi that targets cells expressing functional genes involved in *de novo* uracil biosynthesis such as *URA3* in budding yeast [6], or *ura4* and *ura5* in fission yeast [7,8]. Mutations that inactivate these genes prevent the conversion of 5-FOA into its toxic metabolite, allowing resistant clones to be selected. Typically, such resistance is accompanied by uracil auxotrophy, as *ura3* in budding yeast and *ura4* or *ura5* mutants in *S. pombe* cannot synthesize uracil *de novo* [9]. To understand better how genome instability arises in quiescence compared to growth, we analyzed spontaneous mutation accumulation in both quiescence and growth in *S. pombe* and determined their spectrum using 5-FOA resistance (5-FOA$^R$) [10,11].

In this paper, we describe a new class of 5-FOA$^R$ mutants in *S. pombe* that, in contrast to *ura4* and *ura5* loss-of-function mutants, do not require uracil supplementation for growth. These mutants arise spontaneously and map to a single essential gene. The most likely explanation is that these alleles reduce the conversion of 5-FOA into the toxic intermediate 5-FU without disrupting the endogenous uracil biosynthesis pathway. As a result, they confer resistance to 5-FOA while preserving uracil prototrophy, making them non–counter-selectable with the 5-FOA drug.

## Materials and methods

### *S. pombe* media and general methods

Wild type *S. pombe* strains PB1623 and PB1653 are unswitchable prototrophic derivatives from the h⁻ 972 isolate that are Msmt0 and PD17 at *mat1*, respectively [12–14]. These strains were previously sequenced and shown to contain no mutations in either *ura4*, *ura5* or *ura6* [10,11]. All strains are cultured in YE or YES rich medium, or in MM minimal medium containing glutamate as the source of nitrogen (N) [15]. In MM w/o N glutamate is omitted. All liquid media contain 2% glucose, and solid media are obtained by adding 20 g/L of agar. 5-FOA plates are obtained by adding 1.6 g/L of 5-FOA (Melford) to YE. Phloxine B was added to YPD medium [16] at a final concentration of 5 mg/L [9,16]. *S. pombe* is cultured at 32°C unless stated otherwise. Standard genetic crosses are performed for the complementation tests. Genomic DNA for whole genome sequencing was prepared with the spheroplast methods described in (Murrey et al. Cold Spring Harb Protocol 2016 https://doi.org/10.1101/pdb.prot090985).

### Mutation accumulation in quiescence

The strains are streaked on MM plates and incubated at 32°C until single colonies form. A single colony (~1–2 × 10⁶ cells) is inoculated into 20 mL of MM + 2% glucose and incubated at 32°C with shaking at 180 rpm. When the culture reaches ~4 × 10⁶ cells/mL, cells are harvested by centrifugation at 3,000 rpm for 3 minutes in a 50 mL Falcon tube, washed with 5 mL of nitrogen-free MM (MM w/o N), and transferred into 100 mL of MM w/o N at a starting density of 1 × 10⁶ cells/mL. Cultures are incubated at 28°C with agitation for at least 16 hours to allow the completion of the two divisions required to arrest cells in G0, defining this point as day 1 of quiescence [11].

Viability is assessed at days 1, 2, 3, 4, and 6. At each time point, equal aliquots of the culture are plated on 5-FOA plates at $1–2 \times 10^7$ cells per plate and incubated at 32°C. Resulting 5-FOA$^R$ colonies are picked and analyzed as described below. Mutation frequency at each time point is calculated relative to survival for both 5-FOA$^R$ Ura$^-$ (no growth on MM) and 5-FOA$^R$ Ura$^+$ (growth on MM) clones. The rate of mutation accumulation over time is determined for each experiment, and values are compiled for statistical analysis. A total of 10 independent mutants experiments were performed. The values were analyzed using an "ordinary one-way ANOVA test with GraphPad Prism version 10.6.1 for MacOS, GraphPad Software, Boston, Massachusetts USA, www.graphpad.com".

## Mutation rate in growth

To establish the mutational rate in growth, Luria–Delbrück fluctuation experiments are performed [17]. A prototrophic wild-type strain is streaked on MM to obtain single prototrophic colonies. A single colony is next streaked on YES and incubated until single clones form. Starting from the YES plate, 15 colonies are inoculated each in 2 mL of liquid YE medium supplemented with 125 mg/mL of adenine and incubated at 32°C at 180 rpm. When the cell concentration reaches early stationary phase ($1–2 \times 10^8$ cells/ml), three cultures out of 15 are used to assess viability by plating appropriate dilutions on YES. Cells from all the cultures are collected and resuspended each in 1 mL dH$_2$O + 2% glucose [18] and plated on 5-FOA plates at $1–2 \times 10^7$ cells per plate and incubated at 32°C till 5-FOA$^R$ clones are formed. These clones are next analyzed as described below. Thirteen independent experiments were performed. The mutation rate in each experiment and for both FOA$^R$ Ura$^-$ (no growth on MM) and FOA$^R$ Ura$^+$ (growth on MM) clones is calculated using the software FAL-COR (https://lianglab.brocku.ca/FALCOR/) with a Lea-Coulson Method of the Median [19] or a MSS Maximum Likelihood Method of analysis [20]. The rates were compared using an "ordinary one-way ANOVA test with GraphPad Prism version 10.6.1 for MacOS, GraphPad Software, Boston, Massachusetts USA, www.graphpad.com".

## Integration of wild-type copy of *ura6* at the neutral SPBC31A8.02 locus

Four overlapping fragments encompassing the *ura6* wild-type locus, the hygromycin resistance cassette from pFA6a-hphMX6 (Addgene Plasmid #19342) as well as the 5' and 3' borders of the SPBC31A8.02 locus were amplified by PCR and assembled seamlessly and in the correct order by the Gibson Assembly technique [21]. The resulting PCR fragment was used to transform the identified *ura6* mutants by electroporation. The transformants were first selected for hygromycin resistance and the correct integration was next verified by PCR. All primers used are listed in S1 Table.

## Analysis of 5-FOA$^R$ clones by replica plating

Colonies that form on 5-FOA plates are picked and re-streaked onto fresh 5-FOA plates for validation. Plates are incubated overnight at 32°C and subsequently replica plated onto YES, YPD + Phloxine B, and MM following exactly this sequence. After incubation at 32°C, 5-FOA$^R$ clones are classified as Ura$^-$ (no growth on MM) or Ura$^+$ (growth on MM). 5-FOA$^R$ Ura$^-$ clones are further assigned to either the *ura4* or *ura5* gene using YPD + Phloxine B replicas: *ura4* mutants stain red upon cell death due to Phloxine B uptake, whereas *ura5* mutants remain viable and unstained, as described in [9]. The frequencies of 5-FOA$^R$ Ura$^-$ and 5-FOA$^R$ Ura$^+$ clones are determined and subsequently used to calculate either the mutation rate during growth or the mutation accumulation as a function of time in quiescence.

## Bioinformatic analysis

Whole-genome sequencing was performed by Novogene (UK) on Microbial whole genome libraries using Illumina NovaSeq X Plus Series (PE150). Reads were aligned to the *S. pombe* ASM294v2 reference genome [22] using the Burrows-Wheeler Aligner [23], Picard (http://broadinstitute.github.io/picard), SAMtools [24], and Genome Analysis Toolkit v4.2 [25]. Variants were called against the reference genome using GATK HaplotypeCaller v4.1.4.1. The effect of the

polymorphisms detected was predicted using snpEff [26]. The pipeline used to extract variants was previously described in [11]. Sequencing data were deposited in the NCBI database with SRA PRJNA1272799 and URL https://www.ncbi.nlm.nih.gov/bioproject/?term=PRJNA1272799.

## Results

### 5-FOA resistant colonies can grow in the absence of uracil

We have previously shown that mutations accumulate linearly as a function of time in quiescence that is induced by nitrogen starvation in *S. pombe* [11]. The spontaneous mutational accumulation was established by selecting for 5-fluoroorotic acid (5-FOA) resistant mutants at day 1, 2, 3, and 6 of quiescence (Materials and methods). In *S. pombe*, the 5-FOA$^R$ mutants cluster in *ura4* and *ura5* genes and are unable to grow in the absence of uracil (FOA$^R$ MM$^-$). However, we noticed that some 5-FOA$^R$ resistant mutants were able to grow on minimal medium lacking uracil (FOA$^R$ MM+) (Fig 1A). Unlike mutations in *ura4* or *ura5*, the FOA$^R$ MM$^+$ mutants did not accumulate progressively over time in G0 (Fig 1B). We asked whether these mutants arise only during the first days of quiescence or if they arise also spontaneously during growth. Thus, we performed fluctuation tests to determine the rate of 5-FOA resistance and found that 5-FOA$^R$ MM+ mutants arise spontaneously during growth (Fig 1C). To assess the stability of this phenotype, we examined four 5-FOA$^R$ MM$^+$ clones of each mating type isolated at day 1 of quiescence (PB3636, PB3637, PB3638, PB3639 for M isolated in PB1623; PB3643, PB3644, PB3645, PB3646 for P isolated in PB1653). Several hundred cells from colonies grown on rich medium were plated on the same medium at a density of 100 colonies per plate, then replica-plated first onto MM and next onto 5-FOA, together with wild-type and *ura5* mutant controls. As expected, wild-type cells grew on MM but not on 5-FOA, while *ura5* mutants showed the opposite pattern. In contrast, all eight mutant clones formed colonies on both media, indicating a homogeneous and stable phenotype within colonies. To test inheritance over generations, a single colony of all the strains picked from the non-selective plates of the first round was grown for 20 additional generations, then replated and replica plated as before. In all cases, the ability to grow on both media was retained in the several hundred colonies analyzed (S2 Table), supporting that the phenotype is stably inherited and unlikely to result from an epigenetic change. We next backcrossed PB3643 with the PB1623 wild-type strain and analyzed the segregation of 5-FOA resistance and uracil prototrophy in 38 four-spored tetrads (S1 Fig). As expected, 5-FOA resistance segregates 2:2 while uracil prototrophy segregates 4:0 indicating that 5-FOA resistance always segregates with uracil prototrophy. Together, these results strongly suggest the presence of a novel, stable mechanism of 5-FOA resistance in *S. pombe*. To investigate whether FOA$^R$ MM$^+$ mutants map to a single genetic locus, we next performed complementation tests. A single representative resistant clone from each mating type (parental strains PB3637 and PB3643) was crossed with all four clones of the opposite mating type. Tetrad dissection and analysis of at least seven four-spored asci per cross revealed no 5-FOA sensitive progeny (non-parental ditypes), indicating that all eight mutations are linked to the same locus (Fig 1D), referred to here as "gene X".

### Identification of the locus

To identify the locus responsible for the 5-FOA$^R$ MM$^+$ phenotype, we performed whole-genome sequencing on a complete tetrad from each of the eight crosses previously described, totaling 32 spores. Since each parent (PB3637 and PB3643) is involved in five crosses, we should find ten occurrences of the parental mutations among the 32 genomes (S3 Table). Moreover, these mutations should cluster within a single genomic region, allowing to pinpoint the underlying locus. The result of this analysis revealed that the parental mutations in "gene X" are two non-synonymous missenses (G10S and I62F) in the essential *ura6* gene [27,28] encoding an uridylate kinase involved in the production of UDP (SPCC1795.05c) and well conserved in fungi and mammalian cells (Fig 2A) [29]. Among the six remaining strains, one harbors an allele identical to a parental mutation but arising independently (G10S); two carry mutations affecting the same codon but leading to different amino acid substitutions (S33I, S33N); two contain non-synonymous changes (R39C, S133R); and one possesses an exact in-frame duplication of 12 amino acids (Fig

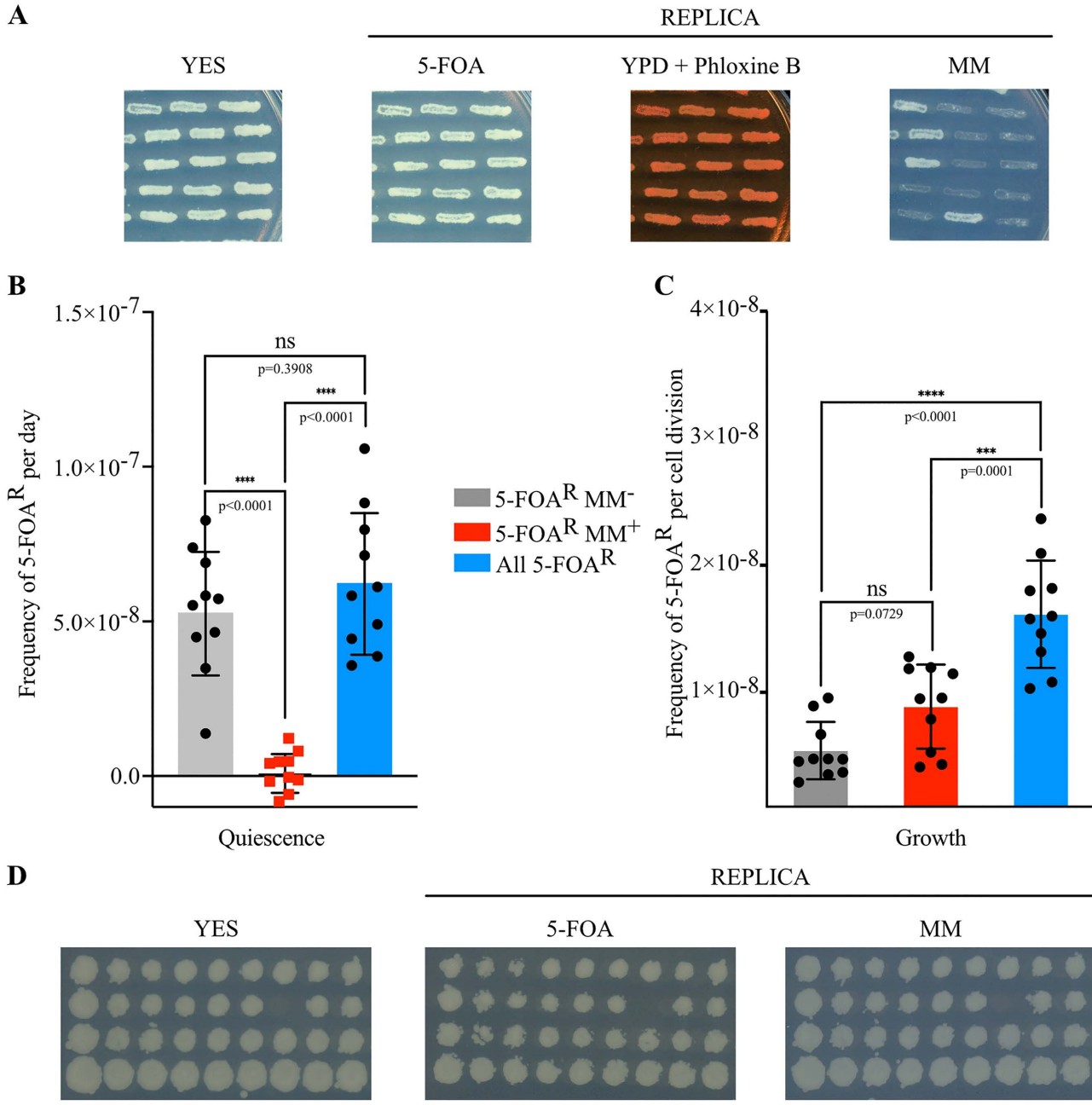

**Fig 1. Identification of 5-FOA^R, uracil-prototrophic colonies.** (A) Example of clones that are 5-FOA resistant at day 1 of quiescence and tested on YES, 5-FOA, YPD + phloxine B to discriminate between *ura4* and *ura5* mutants and MM to identify the Ura⁻ clones and the prototrophic 5-FOA^R MM⁺ mutants. (B) 5-FOA resistant accumulation in G0. Dots represent the slope of mutation accumulation of 10 independent experiments. (C) Mutation rate of 5-FOA^R clones in growth. Dots represent the mutation rate calculated from 13 independent experiments. Statistical analyses were performed using an "ordinary one-way ANOVA" test. (D) Example of tetrads from one cross among the eight performed between the 5-FOA^R MM+ strains (S1 Fig) showing that resistance to 5-FOA is found in every spore in each four-spored tetrad and that this resistance segregates with the ability to grow on MM (MM⁺, parental ditype phenotype).

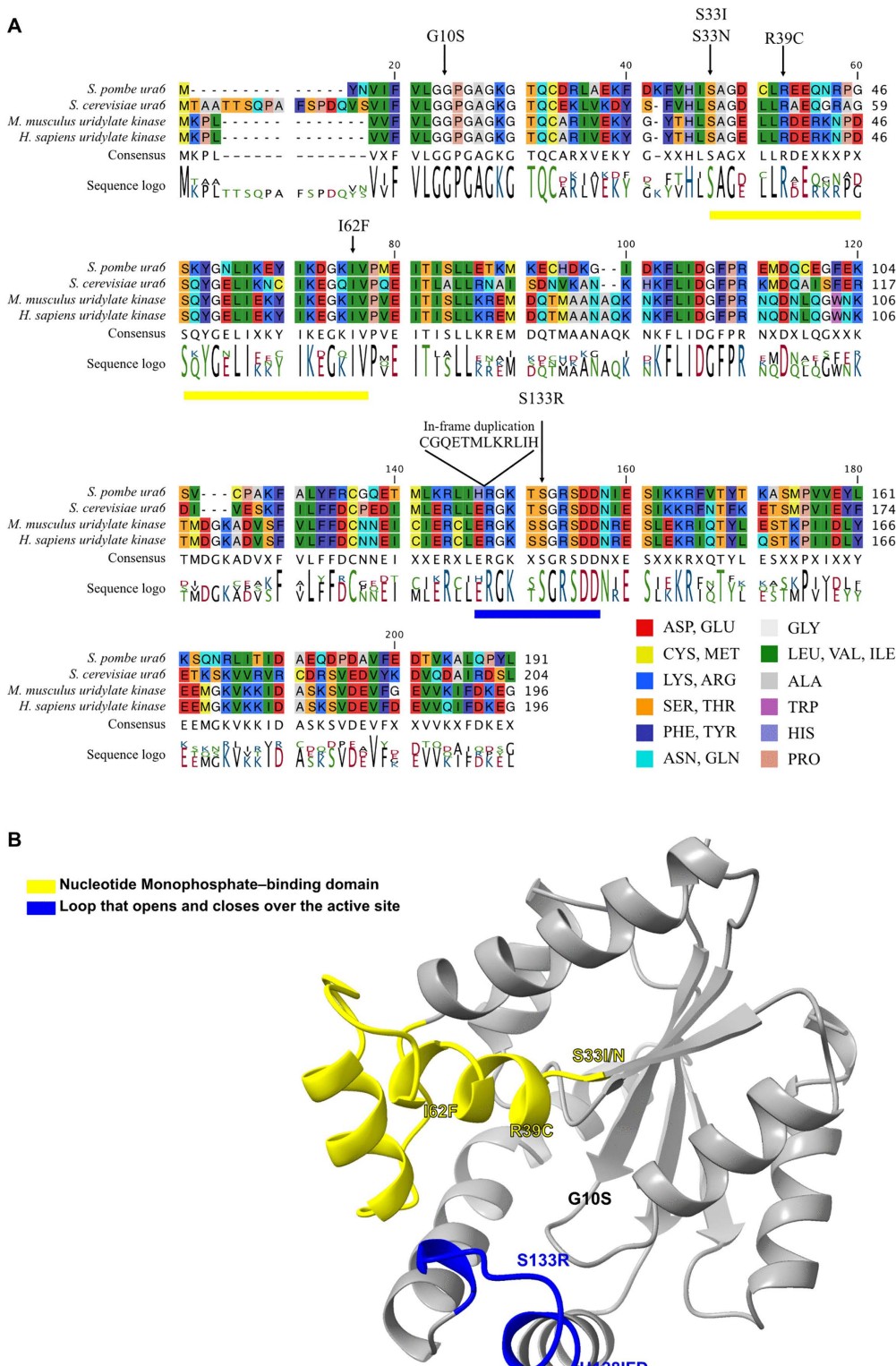

**Fig 2. Alignment of uridylate kinases from different species.** (A) RasMol [31] was used for the graphic representation of the alignment computed with CLC MainWorkbench (QIAGEN Aarhus). The identified mutations are depicted on the sequence alignment. The horizontal yellow and blue bars

under the consensus sequence delineate the nucleotide monophosphate-binding domain and the LID domain of the protein, respectively. (B) Positioning of the found mutations on the AlphaFold generated structure of the *S. pombe* Ura6 protein (UniProt O59771) using the free version of UCSF ChimeraX, version 1.10.1 (2025-07-24) (https://www.rbvi.ucsf.edu/chimerax).

2A and S3 Table). Since *ura6* is essential, it is expected that all recovered mutations are hypomorphic, with either missense or in-frame insertion variants anticipated to have mild impacts on the protein activity by the snpEff program predicting the effects of polymorphisms [26]. In addition, all the mutations that were identified lie in conserved regions of the protein (Fig 2A). The positioning of the mutations on the AlphaFold predicted structure (UniProt O59771) revealed that four mutated positions (S33I/N, R39C and I62F) map to the nucleotide monophosphate-binding domain while two others map to a loop domain described to open and close over the active site to regulate substrate binding, catalysis, or product release [30] (Fig 2B).

Because *ura6* is an essential gene, we first tested whether the mutations caused a temperature-sensitive phenotype. Serial dilutions of each mutant were spotted onto minimal medium (MM) and rich medium alongside the parental wild-type strains, then incubated at 25°C, 30°C, 32°C, 34°C, and 36°C. All mutants formed colonies at all five temperatures, with only a slight delay in colony appearance on MM, most noticeable at the lowest temperatures. The drops spotted on MM at 25, 32 and 36°C are shown in S2 Fig. We confirmed this growth defect by measuring the doubling times of the mutants in MM liquid medium at 25°C and 32°C and by comparing them with the wild-type strains. The results (S4 Table) show that all mutants grow more slowly than wild-type strains PB1623 and PB1653 at both temperatures. PB3636, which carries the in-frame duplication, grows particularly slowly, with a doubling time at 25°C that is more than twice that of the controls. PB3637 and PB3646, both carrying the G10S mutation in an unassigned functional domain of the protein, display identical doubling times, as expected. Plating efficiency was also determined for each mutant and indicates that cell death does not account for the slower growth except for PB3636 at 32°C (S4 Table). Unlike budding yeast, fission yeast undergoes only a transient diploid phase, making it difficult to assess whether mutations are dominant or recessive. To overcome this limitation, we integrated a wild-type copy of the *ura6+* locus into a pseudogene on chromosome II (SPBC31A8.02) and assessed the resulting strains for growth and 5-FOA resistance. In all cases, wild-type growth and 5-FOA sensitivity were fully restored, indicating that the mutations are recessive (S3 Fig).

## Discussion

We identified *ura6* as a novel gene whose mutation confers resistance to 5-FOA while preserving uracil prototrophy. Since *ura6* is an essential gene, all viable *ura6* mutations must retain sufficient activity to support uracil biosynthesis, accounting for their 5-FOA-resistant yet prototrophic phenotype. In agreement with this hypothesis, 6 out of 7 mutations localize to regions important for the catalytic activity of the protein.

We would like to emphasize that *ura6* mutants are included together with *ura4* and *ura5* mutants when calculating the rates and frequencies of total 5-FOA[R] clones. Although *ura6* mutants can grow without uracil and therefore cannot be counter-selected on minimal medium, this does not bias frequency estimates, as such mutations are rare even without counter-selection. Moreover, any *ura6* mutant colony that appears on non-selective medium would be treated as a jackpot event and excluded from the calculation of the rate. Interestingly, the contribution of Ura[+] 5-FOA[R] (*ura6*) mutations to the overall 5-FOA[R] mutation rate in growth is unexpectedly high across 13 independent measurements. Although its mutation rate is not significantly different from the combined rates of *ura4* and *ura5* mutants (Fig 1C), *ura6* accumulates more mutations than either *ura4* or *ura5* individually, despite being an essential gene with a genomic size comparable to that of *ura4* or *ura5*. Moreover, *ura6* mutants should exhibit a biased mutational spectrum with insertions and deletions (INDELs) being underrepresented since they are likely to cause loss of function, which is lethal in an essential gene. Consistent with this, *ura6* mutants do not accumulate linearly in quiescent cells, where INDELs and single-nucleotide variants (SNVs) arise at comparable rates, unlike proliferative conditions, in which SNVs dominate [11,32].

While preparing this manuscript, we came across a bioRxiv preprint reporting the isolation of *S. cerevisiae* mutants in the *URA6* gene that, like in *S. pombe*, confer resistance to 5-FOA while retaining uracil prototrophy [33]. Given the high conservation of uridylate kinase across species and its essential role in pyrimidine metabolism and RNA synthesis, specifically in converting UMP to UDP, it is not surprising that this gene is essential in both yeasts and that some of the alleles we identified in *S. pombe* were also observed in *S. cerevisiae*.

In summary, our findings uncover an alternative 5-FOA resistance mechanism based on hypomorphic mutations in an essential gene and provide new insights into how an essential gene function can be modulated to accommodate toxicity and metabolic activity.

## Supporting information

**S1 Fig. Tetrad dissection of a cross between PB3643 and PB1623.** 38 four-spored tetrads were replica plated onto 5-FOA and MM medium and allowed to grow for two days at 32°C.
(TIF)

**S2 Fig. Drop test to assess thermo-sensitivity.** 3 µL drops of 2-fold dilutions of exponentially growing cultures adjusted at $4 \times 10^5$ cells/mL were spotted on MM medium and incubated at 25°C, 32°C and 36°C for 4 days.
(TIF)

**S3 Fig. Test of dominance and recessiveness.** The wild-type copy of the *ura6 +*gene was inserted into the SPBC31A8.02 pseudogene in the *ura6* mutants identified except for PB3646 that carries the same mutation as PB3637. Around 100 colonies from the parental and corresponding transformed strains were plated onto MM and 5-FOA media and scored after four days at 32°C. All the parental mutants are corrected by the insertion of the wild-type copy of *ura6 +*, indicating that all the mutations are recessive.
(TIF)

**S1 Table. Primers used to integrate a wild-type copy of *ura6+* at the SPBC31A8.02 locus.**
(XLSX)

**S2 Table. Test of the stability of the 5-FOA resistant uracil prototroph phenotype.** PB3636, PB3637, PB3638 and PB3839 are 5-FOA resistant prototrophic cells isolated in the M background, while PB3643, PB3644, PB3645 & PB3646 are 5-FOA resistant prototrophic cells isolated in the P background. PB1623 is a wild-type prototrophic strain in the M background and *ura5* is a 5-FOA resistant mutant in *ura5*. Colonies plated on rich medium (YES) were replica plated onto YES, MM and 5-FOA, and cells growing on any selective medium were counted.
(XLSX)

**S3 Table. Identification of *ura6* mutations in tetrads.** Identification of the mutations in *ura6* by whole genome sequencing of spores coming from the indicated crosses. The M parental strain used to cross with the four P strains is colored in blue while the symmetrical P one is colored in red. Column C indicates the position of the mutation on chromosome 3. Column D indicates the nucleotide in the wild-type sequence of the *ura6⁺* gene that was found mutated. Column E indicates the mutated nucleotide found in the *ura6* mutants. Column F indicates the amino acid in the wild-type proteins while column G shows the change found in the mutant protein. Column H is indicating the position of the mutations in the protein Ura6.
(XLSX)

**S4 Table. Doubling time and plating efficiency of the mutants.** A colony from each mutant strains and one for each wild-type strain were inoculated in MM medium at 25°C and 32°C at $10^6$ cells/mL (Coulter 5000). Growth was followed at different times by measuring cell concentration with a Beckman Coulter Z1. Linear regression in the exponential growth

was used to determine the doubling time. Growing cultures were plated out on three MM plates at 150 colonies per plate at both 25°C and 32°C. Colonies formed after incubation were counted and plating efficiency calculated.
(XLSX)

## Acknowledgments

We thank Artur Scherf for the critical reading of the manuscript.

## Author contributions

**Conceptualization:** Benoît Arcangioli, Stefania Francesconi, Serge Gangloff.

**Data curation:** Stefania Francesconi, Serge Gangloff.

**Formal analysis:** Benoît Arcangioli, Stefania Francesconi, Serge Gangloff.

**Funding acquisition:** Benoît Arcangioli.

**Investigation:** Constance Kowal, Ying Liu, Claire Denis.

**Methodology:** Serge Gangloff.

**Software:** Serge Gangloff.

**Supervision:** Stefania Francesconi, Serge Gangloff.

**Validation:** Stefania Francesconi, Serge Gangloff.

**Writing – original draft:** Benoît Arcangioli, Stefania Francesconi, Serge Gangloff.

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
