## [Decision Letter · Decision Letter 0]

27 Oct 2025

Dear Dr. Gangloff,

Clarity and Reproducibility

The methods and media used in this study are not fully described and insufficient for reproduction by others..The description of the mutant isolation must be described in detail.The meaning of various sections were obscured by lab jargon and a lack of clarity of expression.The references cited contain errors. This must be corrected.

Experimental Issues

The structural predictions for ura6 alleles should be ascertained relative the published and alpha-fold predicted structure of the Ura6 protein1Genetic characteristics of the ura6 mutants (recessive or dominant, temperature sensitivity, etc) need to be competed and described.To generate a ura 6 mutation frequency, Reviewer 2 suggests a direct FOA screen can be used if the study is to be used as an assay for DDR.I concur and the feasibility of a direct approach needs to at least be attempted.In that vein the link of these mutations to genome integrity is indirect and based on the screen. Is it not conceivable that these mutants may be a byproduct of the mutation scheme published in eLife? Have direct assays on these mutations be conducted? If so, describe these studies. If not, these assays must be performed.Both reviewers agreed on the problems in the manuscript but not the remedy. The Academic Editor weighed these comments to generate the list of issues to be addressed above.

We look forward to receiving your revised manuscript.

Kind regards,

Arthur J. Lustig, PhD

Academic Editor

PLOS ONE

**Journal Requirements:**

1. When submitting your revision, we need you to address these additional requirements. Please ensure that your manuscript meets PLOS ONE's style requirements, including those for file naming. The PLOS ONE style templates can be found at https://journals.plos.org/plosone/s/file?id=wjVg/PLOSOne_formatting_sample_main_body.pdf and https://journals.plos.org/plosone/s/file?id=ba62/PLOSOne_formatting_sample_title_authors_affiliations.pdf 2. We note that the grant information you provided in the ‘Funding Information’ and ‘Financial Disclosure’ sections do not match.  When you resubmit, please ensure that you provide the correct grant numbers for the awards you received for your study in the ‘Funding Information’ section. 3. Thank you for stating the following in the Acknowledgments Section of your manuscript: We thank Artur Scherf for support. We note that you have provided funding information that is not currently declared in your Funding Statement. However, funding information should not appear in the Acknowledgments section or other areas of your manuscript. We will only publish funding information present in the Funding Statement section of the online submission form. Please remove any funding-related text from the manuscript and let us know how you would like to update your Funding Statement. Currently, your Funding Statement reads as follows: The author(s) received no specific funding for this work. Please include your amended statements within your cover letter; we will change the online submission form on your behalf. 4. Please note that your Data Availability Statement is currently missing the repository name and/or the DOI/accession number of each dataset OR a direct link to access each database. If your manuscript is accepted for publication, you will be asked to provide these details on a very short timeline. We therefore suggest that you provide this information now, though we will not hold up the peer review process if you are unable. 5. When completing the data availability statement of the submission form, you indicated that you will make your data available on acceptance. We strongly recommend all authors decide on a data sharing plan before acceptance, as the process can be lengthy and hold up publication timelines. Please note that, though access restrictions are acceptable now, your entire data will need to be made freely accessible if your manuscript is accepted for publication. This policy applies to all data except where public deposition would breach compliance with the protocol approved by your research ethics board. If you are unable to adhere to our open data policy, please kindly revise your statement to explain your reasoning and we will seek the editor's input on an exemption. Please be assured that, once you have provided your new statement, the assessment of your exemption will not hold up the peer review process. 6. PLOS ONE now requires that authors provide the original uncropped and unadjusted images underlying all blot or gel results reported in a submission’s figures or Supporting Information files. This policy and the journal’s other requirements for blot/gel reporting and figure preparation are described in detail at https://journals.plos.org/plosone/s/figures#loc-blot-and-gel-reporting-requirements and https://journals.plos.org/plosone/s/figures#loc-preparing-figures-from-image-files. When you submit your revised manuscript, please ensure that your figures adhere fully to these guidelines and provide the original underlying images for all blot or gel data reported in your submission. See the following link for instructions on providing the original image data: https://journals.plos.org/plosone/s/figures#loc-original-images-for-blots-and-gels.  

Reviewers' comments:

**Comments to the Author**

1. Is the manuscript technically sound, and do the data support the conclusions?

Reviewer #1: Partly

Reviewer #2: Partly

2. Has the statistical analysis been performed appropriately and rigorously?

Reviewer #1: N/A

Reviewer #2: I Don't Know

3. Have the authors made all data underlying the findings in their manuscript fully available?

Reviewer #1: Yes

Reviewer #2: Yes

4. Is the manuscript presented in an intelligible fashion and written in standard English?

Reviewer #1: Yes

Reviewer #2: Yes

**Reviewer #1:** This study identifies a novel class of 5-FOA-resistance in S. pombe as having mis-sense mutations in the conserved essential ura6 gene, without affecting growth in the absence of uracil. Aside from the identification of these mutations, there is no other information leading to insight into mechanism.

This is a one-hit wonder with minimal characterization and probably. more suitable for a Micropublication. Are these dominant or recessive? Can they speculate about the domains of the protein affected? What is the mechinism Why have ura6 mutations never been identified previously? (Possibly because everyone expects and screens for ura auxotrophy?)

The authors should at least briefly describe how they performed their selection and determined the results in Fig 1B. There is nothing in the materials and methods that would allow someone to reproduce this. Did they investigate ura6-depenednet FOA resistance in cells NOT in quiescence.

**Reviewer #2:**  Review of PLOS ONE PONE-D-25-45815 Kowal et al. “Hypomorphic mutations in ura6 confer 5-FOA resistance in fission yeast”

Summary:

This short manuscript investigates a subset of 5-FOA-resistant (FOAr) mutations that arose in stationary phase cells in an experimental protocol previously described in and eLife paper by the senior author. Unlike other FOAr mutations, the frequency did not increase with time in stationary phase, occurred in growing cells and FOAr cells did not need exogenous uracil to grow (in marked contrast to the previously known ura4 and ura5 mutations). Using tetrad analysis and whole genome sequencing of several mutants, the authors identify ura6 as the mutated gene. The ura6 gene is essential in S. pombe, so the identified mutations are presumed to be hypomorphs. The mutations are highlighted on an alignment of ura6 orthologs fission and budding yeast and mice and humans. The authors appropriately mention a bioRxiv manuscript for a similar ura6 discovery in S. cerevisiae (which I also read).

Critique:

The discovery of ura6 mutants with an FOAr resistant, Ura+ phenotype is interesting and complementary to the S. cerevisiae work, as it is useful to see that this phenotype is conserved across a large evolutionary distance. However, a criteria for PLOS ONE papers is that the work is complete, and the manuscript has several deficits in the details of the methods, clear descriptions of what was actually done and essential experiments to complete the work and provide an understanding of the mutations. Major concerns include determining if any of the mutants are temperature-sensitive for growth, if ura6 mutants can be directly selected on minimal medium plus FOA and better representation of where the mutants are with respect to active domains of the ura6 protein or mapping them onto the protein structure. My concerns about the manuscript for publishing in PLOS ONE are enumerated below.

1. The link to genome integrity in the manuscript is misleading as mechanisms of genome integrity are never assayed (effects of DNA replication, DNA repair, etc.). It appears that the authors were following a protocol from the senior author’s 2017 eLife paper and these mutations were later identified as a byproduct and are now being characterized. However, the methods does not describe how these experiments were done and the fact that they were published is not sufficient because the reader does not know how the experiments were done. This point is important, as anyone who wishes to use this discovery as an assay needs a complete description of what was done to replicate it in their own lab to assay mutants (e.g. in DNA replication and DNA repair).

2. The methods section is incomplete. The authors need a complete description of how the ura6 mutants were isolated in quiescent cells and proliferative cells. I can find no details in the manuscript for what was exactly done, and the proliferative cell experiments are distinct from the 2017 paper.

3. The media descriptions are incomplete. YES (spelled YEs by the authors) have supplements that can vary between labs or experiments, and the strains are prototrophic in many experiments, the supplements and concentrations should be explicitly stated.

4. Pg. 5 lines 77 to 81. The description of the mutant isolation and the previous methods is incomplete and confusing. When S. pombe cells are starved for nitrogen to induce quiescence, cells can grow for one or two divisions before arresting. How many divisions take place in these experiments? How does this define the accumulation of ura4, 5 and 6 mutations? Do the frequency of mutant cells increase as the days in quiescence increases, and if so how much (i.e. are the numbers of ura4, ura5 and ura6 mutants produced on different days of quiescence known)? A better description is needed. As stated in point 2, how were the proliferative experiments done? I can see no way to get a reasonable estimate as to the frequency at which the ura6 mutations arise. From the data presented, it could be that the ura6 FOAr mutants are so rare that changes in the number is so slow that an increase cannot be detected in these experiments.

5. Pg 5 lines 82 to 84 – the replica plating experiments are not really described. The methods should have some mention for the many replica plating experiments in the manuscript for the general protocol: specifically, how many colonies per plate were used in assays and, in the Results, how many plates. I cannot tell if the work was done poorly (too many colonies per plate to count the replica plates well) or carefully (<200 colonies per plate, with control replica plates to monitor transfer for colonies, e.g. first replica plate is discarded, second is the test plate and third plate is one where all cells can grow).

6. Pg. 5 lines 87-89. I cannot follow the experimental details here – was the non-selective growth in liquid or a single colony on plates? My concerns in point 5 above apply here as well, and could be answered by a more complete methods section.

7. Pg. 6 lines 107-109. The legend for Figure 1C needs to describe that the cross is between two resistance isolates. Also, the conclusion that “5-FOA resistance co-segregates with the ability to grow on MM” is not supported since both parents lack auxotrophic markers and can grow on MM (also, aa cytoplasmic factor that allows growth would produce a similar result). Figure S1 is more supportive of this single locus conclusion, although the figure is not great because the FOA sensitive colonies show some growth. Any comments regarding co-segregation of growth on MM requires a cross with a linked auxotrophic marker so the FOA-resistance and growth on MM clearly segregate from the FOA-sensitive auxotrophy.

8. Pg. 6 lines 110-118. This section was very confusing and filled with lab jargon. The authors state one complete tetrad from 8 different crosses was sequenced (line 112), but line 115 refers to 4 crosses, not 8. Line 115 also states in “the symmetrical cross”, a term that I am not familiar with despite decades of work in classical and molecular yeast genetics. The phrase “the roles of the two parental strains” is not clear (what does “roles” mean) and the authors need to be more explicit in both cases. Finally, I could not follow the logic that a mutation segregating 2:2 in 32 spores from 8 tetrads should be present 10 times in 32 genomes instead of 16. This section needs to be clarified.

9. Pg. 6 lines 130-132 and Figure 2. The alignment of orthologs is not strongly informative except for conservation and could be in the supplemental material. The product of the ura6 gene, uridylate kinase, has been studied and enzymatic domains identified, listed in UniProt (enter O59771) and Pombase. Where are the authors’ mutations with respect to the most important regions of this essential protein? The structure of the S. cerevisiae enzyme was solved in 1994 and the predicted structures from AlphaFold for the S. pombe enzyme is quite similar. It should be easy for the authors to find a structural biologist at their institution that can get the AlphaFold structure, highlight the active or important domains and color the mutations the authors have found. A student with a passing knowledge of PyMOL or ChimeraX could do this in less than hour. Such a figure would set the authors’ work apart from the S. cerevisiae bioRxiv paper which has not made use of the structural data.

10. Results section-missing experiments – Temperature-sensitivity. The ura6 gene is essential, so some of the mutants may be temperature-sensitive. Conditional growth phenotypes must be tested for a complete analysis.

11. Results section-missing experiments – Testing for direct selection for ura6 FOAr mutants. The standard FOA medium for S. pombe is a minimal medium plus FOA and supplements (see https://dornsife.usc.edu/pombenet/drugs/). Consequently, one should be able to directly select for ura6-hypomorphs in a growing population using minimal medium plus FOA and lacking uracil, especially since the authors state on p. 5 line 80 that the mutations occur during proliferative growth. For a complete manuscript, this experiment needs to be tried to demonstrate the possibility and determine if ura6 FOAr mutations (missense, duplications) are reasonable assay. The authors could present an approximate frequency of growing cells and use PCR and sequencing to show the presence of ura6 mutations. A “reconstruction experiment” where FOAr ura6 hypomorphs are diluted into a wild type culture in a ratio of 1:10000 would also be helpful to determine the efficiency of this approach. This experiment is important if the phenotype of ura6 hypomorphs is to be used in assays for mutation frequencies or rates in mutants of DNA repair or replication.

12. Pg. 8 lines 145-147. The linear accumulation of mutants in quiescent cells is cited again, and compounds the issues of not explaining this protocol at the beginning of the results section (point 4 above).

13. Pg 8 line 158. Acknowledgments. Was this work funded by a grant? How were the workers supported? What kind support did Artur Scherf provide? Technical? Financial? Lab space? Explicit information similar to other PLOS ONE articles has to be provided.

14. Pg. 10 line 207-209. The reference appears to be wrong. Check PMID: 22728672. I have no idea how this reference can support the statement on page 7 line 128 and whether any reference is needed to make that point.

15. Pg. 11 Tables S1 and S2 legends. The meanings of the column headings need to be described in the legend, especially for Table S2.

**Do you want your identity to be public for this peer review?** For information about this choice, including consent withdrawal, please see our Privacy Policy

Reviewer #1: No

Reviewer #2: No

---

## [Author Response · Author response to Decision Letter 1]

9 Dec 2025

Point to point answers to the Academic Editor’s request

Clarity and Reproducibility

• The methods and media used in this study are not fully described and insufficient for reproduction by others.

Methods and media have been extensively revised to improve reproduction of experiments by others. We have added detailed descriptions of the experiments to establish mutation accumulation in quiescence and to determine mutation rate in growth by selecting for 5-FOA resistant clones. The analysis of the 5-FOAR clones by replica plating has also been clearly described in the methods.

• The description of the mutant isolation must be described in detail.

In the revised Results section, we describe in greater detail how we isolated a new class of spontaneous mutants that display 5-FOA resistance while remaining uracil prototrophs. These 5-FOA-resistant MM⁺ clones emerged during experiments on mutation accumulation in quiescent cells and were also detected in control experiments with proliferating cells.

• The meaning of various sections were obscured by lab jargon and a lack of clarity of expression.

We eliminated lab-specific terminology and substituted it with clearer and more explicit language.

• The references cited contain errors. This must be corrected.

We verified all references but did not find any errors.

Reviewer 2 notes in point 14 that the reference cited on page 10, lines 207–209 (“set of gene deletions in the fission yeast Schizosaccharomyces pombe. Nat Biotechnol. 2010 Jun;28(6):617–23”) appears incorrect and therefore does not support the statement on page 7, line 128 (“The identified mutations are depicted on the sequence alignment.”). The reviewer suggests checking PMID: 22728672, which corresponds to our reference 21 and is already correctly cited in the “Bioinformatic Analysis” section on page 4, line 67.

Experimental Issues

• The structural predictions for ura6 alleles should be ascertained relative the published and alpha-fold predicted structure of the Ura6 protein1

We have included in the revised version the predicted AlphaFold structure of Ura6 (UniProt O59771) and mapped the mutations that we have identified. As described in the revised manuscript, four of the mutated residues (S33I/N, R39C, and I62F) localize to the nucleotide monophosphate–binding domain, while the remaining two reside in a loop region that is proposed to open and close over the active site to regulate substrate binding, catalysis, or product release.

• Genetic characteristics of the ura6 mutants (recessive or dominant, temperature sensitivity, etc) need to be competed and described.

We have characterized the ura6 alleles for their thermo-sensitivity (drop test, growth curves and plating efficiency) and included the results in the S2 figure and S3 Table.

We could not determine whether the mutations were dominant or recessive because, unlike budding yeast, which alternates between haploid and diploid states, fission yeast is almost exclusively haploid. Its diploid phase is transient, as cells rapidly enter meiosis. The more stable diploids that can be generated (M/M or P/P), either by protoplast fusion or through ade6-M210/ade6-M216 intragenic complementation, are physiologically compromised and display heightened drug sensitivity. In addition, the lack of centromeric plasmids in fission yeast further restricts the possibility of assessing dominance or recessivity in a haploid organism.

• To generate a ura6 mutation frequency, Reviewer 2 suggests a direct FOA screen can be used if the study is to be used as an assay for DDR.I concur and the feasibility of a direct approach needs to at least be attempted.

In the revised version, we now include the direct 5-FOA screening data in Fig. 1C. The figure presents results from 13 independent measurements of the mutation rate at the ura6 locus (5-FOAᴿ MM⁺) in growth compared with ura4 and ura5 (5-FOAᴿ MM⁻). These results are discussed in the Discussion section and were obtained through fluctuation test experiments, the protocol for which is described in detail in the Materials and Methods section.

• In that vein the link of these mutations to genome integrity is indirect and based on the screen. Is it not conceivable that these mutants may be a byproduct of the mutation scheme published in eLife? Have direct assays on these mutations be conducted? If so, describe these studies. If not, these assays must be performed.

In the revised version, we clarify that the link to genome instability lies in the use of ura6 mutants to estimate the rate of mutagenesis during growth, with mutagenesis serving as a proxy for genome instability. However, ura6 mutants introduce a bias when analyzing mutation spectra: insertions and deletions are likely underrepresented because such events typically disrupt gene function and are lethal in an essential gene. Consistent with this, ura6 mutants do not accumulate linearly in quiescent cells, where insertions, deletions, and single-nucleotide variants arise at comparable rates, unlike in proliferating conditions where SNVs predominate.

---

## [Decision Letter · Decision Letter 1]

4 Jan 2026

Dear Dr. Gangloff,

Thank you for submitting your manuscript to PLOS ONE. After careful consideration, we feel that it has merit but does not fully meet PLOS ONE’s publication criteria as it currently stands. Therefore, we invite you to submit a revised version of the manuscript that addresses the points raised during the review process.

**The one major issue that remains is a test of dominance and recessiveness. This can be done by integration into a haploid strain as suggested by Reviewer 1.**
**This experiment needs to be attempted prior to publication. If there are reasons why this cannot be completed please address the editor through the PLOS One portal.**

We look forward to receiving your revised manuscript.

Kind regards,

Arthur J. Lustig, PhD

Academic Editor

PLOS One

Journal Requirements:

Reviewers' comments:

Reviewer's Responses to Questions

**Comments to the Author**

Reviewer #1: (No Response)

Reviewer #2: (No Response)

2. Is the manuscript technically sound, and do the data support the conclusions?

Reviewer #1: Partly

Reviewer #2: Yes

3. Has the statistical analysis been performed appropriately and rigorously?

Reviewer #1: Yes

Reviewer #2: Yes

4. Have the authors made all data underlying the findings in their manuscript fully available?

Reviewer #1: Yes

Reviewer #2: Yes

5. Is the manuscript presented in an intelligible fashion and written in standard English?

Reviewer #1: Yes

Reviewer #2: Yes

Reviewer #1: The authors have improved their MS to address several issues. This study shows a distinct class of spontaneous FOA-R mutation occurs in the ura6 gene that does not confer ura auxotrophy, and now shows this is not limited to quiescence. There is no obvious link to any genome integrity defects.

it is certainly possible to test dominance/recessiveness in a stable diploid (any changes in drug sensitivity are easily adjusted/controlled for).

Reviewer #2: Review of PONE-D-23-45815-R1 Kowall et al. “Hypomorphic mutations in ura6 confer 5-FOA resistance in fission yeast”

The revised manuscript addresses many of my previous concerns.

On the positive side, the performance of the fluctuation tests is commendable. The ability to select ura6 mutations directly on FOA medium lacking uracil was not tested but can be let go.

A few concerns remain that the authors should note or correct.

1. Fig. S1 still uses YEs instead of the standard YES and should be corrected.

2. The AlphaFold program is incorrectly written as alpha-fold (page 9 line 186, page 10 line 210). The methods should cite the website used for the structure (apparently Uniprot?) and the date it was accessed.

It is standard to cite he software used to color the structure. If done at the website, the methods should state that.

3. Dominance vs. recessive nature of the ura6 mutations.

The authors state that they could not determine dominance versus recessivity of the alleles because the S. pombe stable diploids “are physiologically compromised and display heightened drug sensitivity. In addition, the lack of centromeric plasmids in fission yeast further restricts the possibility of assessing dominance or recessivity in a haploid organism.”

I am willing to let the point go. However, the authors could have answered this question by integrating one of their ura6 FOAr mutants at another locus (e.g. an intragenic region) and tested for FOAr in the presence of a wild type ura6+ gene.

The dominance-recessive test would validate the assumption that it is the hypomorphic nature of the enzyme that causes the phenotype.

4. Funding:

The editor needs to clarify whether institutional or government support that is not a grant needs to be acknowledged. Does the senior author get a salary and are the lab members supported by the Pasteur Institute or government? US institutional support normally has to be declared (i.e. internal grants, accounts for salary support), so I would expect similar rules for French or other researchers. What is the position of PLOS on this point?

5 The authors need to check and correct their reference list.

Reference 21 from 2014 cited by the authors in the original submission and the revised submission does not exist in PubMed. The authors cite:

Cingolani P, Platts A, Wang LL, Coon M, Nguyen T, Wang L, et al. A program for

annotating and predicting the effects of single nucleotide polymorphisms, SnpEff. Fly. 2014

Oct 27;6(2):80–92.

However, searching with the title or the first author’s name “Cingolani P“ only yields the 2012 reference with a similar name:

Fly (Austin)

2012 Apr-Jun;6(2):80-92. doi: 10.4161/fly.19695.

A program for annotating and predicting the effects of single nucleotide polymorphisms, SnpEff: SNPs in the genome of Drosophila melanogaster strain w1118; iso-2; iso-3

Pablo Cingolani 1 , Adrian Platts, Le Lily Wang, Melissa Coon, Tung Nguyen, Luan Wang, Susan J Land, Xiangyi Lu, Douglas M Ruden

I do not see how the authors can claim that the PMID: 22728672 for a 2012 reference that I gave was the same as the 2014 reference they cite in both the original and revised versions.

The authors’ response “The reviewer suggests checking PMID: 22728672, which corresponds to our reference 21 and is already correctly cited in the “Bioinformatic Analysis” section on page 4, line 67” is wrong and makes one question how well they actually checked the accuracy of their bibliography.

Citations are a reflection of how we treat our scientific colleagues and need to be accurate. The authors should take this point seriously and check their references again.

**Do you want your identity to be public for this peer review?** For information about this choice, including consent withdrawal, please see our Privacy Policy

Reviewer #1: No

Reviewer #2: No

---

## [Author Response · Author response to Decision Letter 2]

13 Feb 2026

Point to point answers to the Reviewers

Reviewer #1: The authors have improved their MS to address several issues. This study shows a distinct class of spontaneous FOA-R mutation occurs in the ura6 gene that does not confer ura auxotrophy, and now shows this is not limited to quiescence. There is no obvious link to any genome integrity defects.

it is certainly possible to test dominance/recessiveness in a stable diploid (any changes in drug sensitivity are easily adjusted/controlled for).

• We have addressed the dominance/recessiveness of the ura6 alleles by integrating a wild-type copy of ura6+ in a pseudogene that was previously shown in the lab to have no detectable phenotype. The method is described lines 99-106 and in the legend to the S3 Figure (353-358) as well as the primers used to make the construction (S1 Table and 359-360). The results are reported in the result section (lines 212-217). A reference for the Gibson Assembly technique is provided lines 313-314.

Reviewer #2: Review of PONE-D-23-45815-R1 Kowall et al. “Hypomorphic mutations in ura6 confer 5-FOA resistance in fission yeast”

The revised manuscript addresses many of my previous concerns.

On the positive side, the performance of the fluctuation tests is commendable. The ability to select ura6 mutations directly on FOA medium lacking uracil was not tested but can be let go.

A few concerns remain that the authors should note or correct.

1. Fig. S1 still uses YEs instead of the standard YES and should be corrected.

• We corrected the S1 Figure accordingly.

2. The AlphaFold program is incorrectly written as alpha-fold (page 9 line 186, page 10 line 210). The methods should cite the website used for the structure (apparently Uniprot?) and the date it was accessed.

It is standard to cite he software used to color the structure. If done at the website, the methods should state that.

• We corrected the spelling of AlphaFold as requested (lines 194, 223). The source (UniProt, previously referenced in line 194) of the structure is specified again in the legend of Figure 2 (line 224) and the software used to color the structure is provided in the legend (lines 223-225).

3. Dominance vs. recessive nature of the ura6 mutations.

The authors state that they could not determine dominance versus recessivity of the alleles because the S. pombe stable diploids “are physiologically compromised and display heightened drug sensitivity. In addition, the lack of centromeric plasmids in fission yeast further restricts the possibility of assessing dominance or recessivity in a haploid organism.”

I am willing to let the point go. However, the authors could have answered this question by integrating one of their ura6 FOAr mutants at another locus (e.g. an intragenic region) and tested for FOAr in the presence of a wild type ura6+ gene.

The dominance-recessive test would validate the assumption that it is the hypomorphic nature of the enzyme that causes the phenotype.

• We have addressed the dominance/recessiveness of the ura6 alleles by integrating a wild-type copy of ura6+ in a pseudogene that was previously shown in the lab to have no detectable phenotype. The method is described lines 99-106 and in the legend to the S3 Figure (353-358) as well as the primers used to make the construction (S1 Table and 359-360). The results are reported in the result section (lines 212-217). A reference for the Gibson Assembly technique is provided lines 313-314.

4. Funding:

The editor needs to clarify whether institutional or government support that is not a grant needs to be acknowledged. Does the senior author get a salary and are the lab members supported by the Pasteur Institute or government? US institutional support normally has to be declared (i.e. internal grants, accounts for salary support), so I would expect similar rules for French or other researchers. What is the position of PLOS on this point?

• To clarify this point, the salaries of SG, SF and CD are provided by the CNRS (Centre National de la Recherche Scientifique), of BA by the Pasteur Institute, of CK by the Sorbonne Université and that of YL by Changchun Institute of Biological Products Co., Ltd. No specific grant was attributed to this study, but it was supported by recurrent funding from CNRS and Institut Pasteur.

5 The authors need to check and correct their reference list.

Reference 21 from 2014 cited by the authors in the original submission and the revised submission does not exist in PubMed. The authors cite:

Cingolani P, Platts A, Wang LL, Coon M, Nguyen T, Wang L, et al. A program for

annotating and predicting the effects of single nucleotide polymorphisms, SnpEff. Fly. 2014

Oct 27;6(2):80–92.

However, searching with the title or the first author’s name “Cingolani P“ only yields the 2012 reference with a similar name:

Fly (Austin)

2012 Apr-Jun;6(2):80-92. doi: 10.4161/fly.19695.

A program for annotating and predicting the effects of single nucleotide polymorphisms, SnpEff: SNPs in the genome of Drosophila melanogaster strain w1118; iso-2; iso-3

Pablo Cingolani 1 , Adrian Platts, Le Lily Wang, Melissa Coon, Tung Nguyen, Luan Wang, Susan J Land, Xiangyi Lu, Douglas M Ruden

I do not see how the authors can claim that the PMID: 22728672 for a 2012 reference that I gave was the same as the 2014 reference they cite in both the original and revised versions.

The authors’ response “The reviewer suggests checking PMID: 22728672, which corresponds to our reference 21 and is already correctly cited in the “Bioinformatic Analysis” section on page 4, line 67” is wrong and makes one question how well they actually checked the accuracy of their bibliography.

Citations are a reflection of how we treat our scientific colleagues and need to be accurate. The authors should take this point seriously and check their references again.

• Unfortunately, a mistake was present in our database on the year of this publication and has been rectified in our manuscript (line 324).

---

## [Editor Report · Decision Letter 2]

16 Feb 2026

Hypomorphic mutations in ura6 confer 5-FOA resistance in fission yeast

PONE-D-25-45815R2

Dear Dr. Gangloff,

We’re pleased to inform you that your manuscript has been judged scientifically suitable for publication and will be formally accepted for publication once it meets all outstanding technical requirements.

Kind regards,

Arthur J. Lustig, PhD

Academic Editor

PLOS One
---

## [Editor Report · Acceptance letter]

PONE-D-25-45815R2

PLOS One

Dear Dr. Gangloff,

I'm pleased to inform you that your manuscript has been deemed suitable for publication in PLOS One. Congratulations! Your manuscript is now being handed over to our production team.

Kind regards,

on behalf of

Dr. Arthur J. Lustig

Academic Editor

PLOS One